# AMAP: Automatic Multi-head Attention Pruning by Similarity-based Pruning Indicator

## Abstract

Despite the strong performance of Transformers, quadratic computation complexity of self-attention presents challenges in applying them to vision tasks. Linear attention reduces this complexity from quadratic to linear, offering a strong computation-performance trade-off. To further optimize this, automatic pruning is an effective method to find a structure that maximizes performance within a target resource through training without any heuristic approaches. However, directly applying it to multi-head attention is not straightforward due to channel mismatch. In this paper, we propose an automatic pruning method to deal with this problem. Different from existing methods that rely solely on training without any prior knowledge, we integrate channel similarity-based weights into the pruning indicator to preserve the more informative channels within each head. Then, we adjust the pruning indicator to enforce that channels are removed evenly across all heads, thereby avoiding any channel mismatch. We incorporate a reweight module to mitigate information loss due to channel removal and introduce an effective pruning indicator initialization for linear attention, based on the attention differences between the original structure and each channel. By applying our pruning method to the FLattenTransformer on ImageNet-1K, which incorporates original and linear attention mechanisms, we achieve a 30% reduction of FLOPs in a near lossless manner. It also has 1.96% of accuracy gain over the DeiT-B model while reducing FLOPs by 37%, and 1.05% accuracy increase over the Swin-B model with a 10% reduction in FLOPs as well. The proposed method outperforms previous state-of-the-art efficient models and the recent pruning methods.

## 1 Introduction

Transformer has achieved remarkable success in various computer vision tasks based on attention mechanisms that effectively capture long-range dependencies. The attention module generates an attention map by utilizing the query $Q \in \mathbb{R}^{N \times C}$ and key $K \in \mathbb{R}^{N \times C}$ to extract the relationships between tokens, and then projects the value $V \in \mathbb{R}^{N \times C}$ to obtain a feature map with global information. It requires a computational cost of $\mathcal{O}(N^2C)$, which is quadratic with respect to the number of tokens $N$. Despite providing excellent performance, the quadratic complexity with respect to $N$ poses significant challenges for deployment on mobile and edge devices.

Recent research attempts to mitigate this issue by designing efficient transformers. Some approaches propose network architectures that limit the number of tokens (Wang et al. (2021); Liu et al. (2021); Hassani et al. (2023)). By reducing the number of tokens, they cut down computation cost while trying to maintain the performance. However, this results in limitations of the receptive field, constraining the capture of global dependencies. Other approaches propose a new attention mechanism that can replace original attention (Kitaev et al. (2020); Shaker et al. (2023); Han et al. (2023)). Linear attention is a method that maintains the ability to capture long-range dependencies while reducing computational complexity linearly. They first approximate Softmax function by replacing it with a simple activation or a tailored function. By changing the computation order from $(Q \cdot K^T) \cdot V$[1] to $Q \cdot (K^T \cdot V)$[2], they obtain an attention mechanism with computational complexity of $\mathcal{O}(NC^2)$, which is quadratic with respect to channels instead of tokens.

---

[1]Original attention mechanism

[2]Linear attention mechanism

Another strategy to improve the network efficiency is network pruning. It aims to achieve a lightweight network by removing redundant parts from the existing network while minimizing performance degradation. This enables the user to adaptively allocate resources based on their requirements, resulting in a network with computational costs suited to target devices. Human knowledge-based analysis is performed on each element such as gradient, Hessian distribution, and filter properties to measure the extent of redundancy (Gao et al. (2023); Yu et al. (2022b); Yang et al. (2023)). These pruning approaches can be non-optimal, as all elements are deterministically removed based on specific metrics. To tackle this, some approaches find the efficient network structure during the learning process. Automatic pruning involves a hyper-network or pruning indicator with learnable parameters to the target network (Xiao et al. (2019); Li et al. (2020; 2022)). It finds the configuration for each layer that

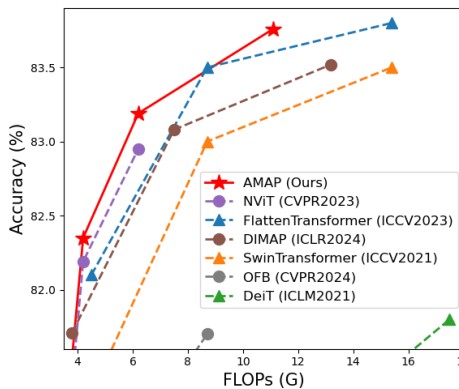

Figure 1: Comparison of other networks with our method. The results demonstrate that the proposed method outperforms not only vision transformer variants ($\triangle$), but also efficient networks using pruning methods ($\bigcirc$).

can maximize performance within a limited resource during the learning process. This eliminates the need for hand-crafted designs tailored to specific networks.

Despite the strength of automatic pruning, applying it directly to Transformer is not trivial. The original attention mechanism is structured with multiple heads, as shown in Fig. 2(a). It enables the model to collectively attend to information from various representation subspaces at different positions (Vaswani et al. (2017)). Fig. 2(b) illustrates the issues when applying automatic pruning without considering multi-head attention. Removing all channels from a specific head eliminates its representation subspace, which restricts the range of features that can be extracted. Consequently, this leads to a notable reduction in the expression capacity compared to the original network. It can induce mixed channels from different representation subspaces, resulting in the extraction of features that are entirely different from those of the original network. We refer to these issues as the *channel mismatch problem.*

Most Transformer pruning research focuses on the original attention mechanism used in DeiT or SwinTransformer. Since linear attention has a computation complexity of $\mathcal{O}(NC^2)$, however, pruning the same number of channels results in a greater reduction in computational cost compared to the original attention. Based on this advantage, we propose an *automatic multi-head attention pruning method* that can be effectively applied to *both original and linear attention mechanisms*. As in previous approaches (Xiao et al. (2019); Li et al. (2022); Lee et al. (2024)), we apply a pruning indicator to identify and eliminate redundant elements through a learning process. Existing automatic pruning methods rely solely on training without considering the expression capacity of the original network. In contrast, *we assign weights according to head-wise similarity when training the pruning indicator to retain the most informative factors for each head.* This approach ensures that the representation subspaces closely resemble the original attention mechanism, minimizing the loss in expression capacity. To address the channel mismatch issue that arises during reconfiguration as in Fig. 2(b), we introduce a pruning indicator adjustment process, which involves head-wise ranking followed by rank-wise averaging. It handle the problem by balancing the pruning ratio for each head, as shown in Fig. 2(c). Our approach leverages the similarity between channels during the training of the pruning indicator, enabling the representation of removed channels through a combination of the remaining ones. Consequently, we compensate for the information loss by incorporating a reweighting module to adjust the scale of the remaining channels. Additionally, we introduce a method to initialize the pruning indicator for linear attention, ensuring that channel removal is not excessive.

We apply it to FlattenTransformer (Han et al. (2023)), a model that integrates both attention mechanisms with state-of-the-art performance, to demonstrate the effectiveness of our proposed approach. On the ImageNet-1K benchmark (Deng et al. (2009)), our approach enables a **30%** reduction in the computational complexity of FLatten-Swin-B in a near-lossless manner. This results in a **1.96%** of accuracy gain over the Deit-B model while reducing FLOPs by **37%**. Our method achieves up to a **1.05%** performance improvement over the SwinTransformer (Liu et al. (2021)) with lower

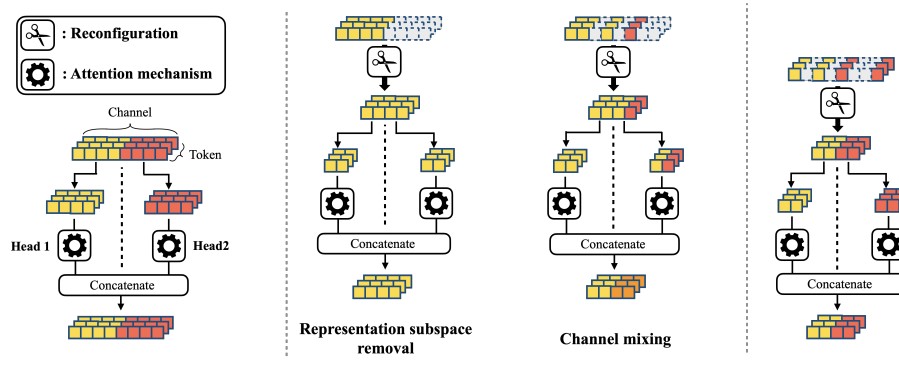

Figure 2: **Problems arising when multi-head is not considered.** (a) Multi-head attention is applied to each head, forming different representation subspaces. (b) Without considering multi-head, reconfiguration leads to channel mismatch problems, where representation subspaces are entirely removed, or get mixed, significantly reducing expression capacity. (c) Our method addresses these problems through an automatic pruning method that takes multi-head into account.

FLOPs. We also compare our approach against efficient networks using pruning methods. As shown in Fig. 1, our method shows higher accuracy at the similar FLOPs compared with previous pruning methods such as NViT (Yang et al. (2023)) and DIMAP (He & Zhou (2024)). To demonstrate the effectiveness across different architectures, we apply our method to the SwinTransformer, achieving better performance than SPViT (He et al. (2024)) with similar FLOPs.

The main contribution of our works are summarized as follows:

(1) We apply an automatic pruning method to find a Transformer structure that enhances performance for both original and linear attention mechanisms within the target computational complexity. (2) To address multi-head attention, we propose a similarity-based pruning indicator that maintains the same proportion across all heads while considering the channel-wise similarity. (3) Experiments show that our method outperforms state-of-the-art efficient transformers.

## 2 RELATED WORKS

### 2.1 EFFICIENT VISION TRANSFORMER

Transformer (Vaswani et al. (2017)) exhibits exceptional performance in the NLP field due to their ability to effectively capture long-range dependencies. The Vision Transformer (ViT) has successfully adapted these models for image classification, achieving outstanding performance (Dosovitskiy et al. (2021)). However, the quadratic computational complexity of the original attention module in Transformers poses challenges for various vision applications. Several methods address this concern by limiting the number of tokens. Pyramid Vision Transformer (PvT) (Wang et al. (2021)) progressively limits the number of tokens using spatial reduction attention, which controls the feature map size in the patch embedding. Deformable Attention Transformer (DAT) (Xia et al. (2022)) selects the positions of keys and values in a data-dependent manner, considering only attentive tokens. SwinTransformer (Liu et al. (2021)) divides the input into windows and performs original attention within these limited regions. Neighborhood Attention Transformer (NAT) (Hassani et al. (2023)) localizes original attention by considering only the nearest neighboring pixels related to the query. Other researches improve efficiency by integrating convolution operations into transformer models. Convolutional vision Transformer (CvT) (Wu et al. (2021)) uses convolutional projections instead of linear projections to control efficiency. CMT (Guo et al. (2022)) proposes a hybrid network of transformers and convolutions, achieving a better trade-off between accuracy and efficiency.

Another approach to achieve efficient vision transformers is to approximate the original attention with linear complexity operations. EdgeNeXt (Maaz et al. (2022)) applies transposed attention along the channel dimension instead of the spatial dimension, achieving linear complexity with respect to tokens. Reformer (Kitaev et al. (2020)) replaces the dot-product operation with locality-sensitive

hashing, reducing the computation to $\mathcal{O}(n \log n)$. LinFormer (Wang et al. (2020)) approximates the original attention matrix using low-rank matrix factorization, achieving linear complexity. Swift-Former (Shaker et al. (2023)) demonstrates that key-value interaction can be replaced with a linear layer without performance degradation, effectively reducing computation. A different line of research involves changing the order of original attention computations to achieve a complexity of $\mathcal{O}(NC^2)$. This requires effectively replacing Softmax. CosFormer (Qin et al. (2022)) replaces Softmax with ReLU activation and cosine-based distance re-weighing. SOftmax-Free Transformer (SOFT) (Lu et al. (2021)) uses a Gaussian kernel function to replace Softmax and achieves linear complexity through low-rank decomposition. Castling-ViT (You et al. (2023)) extracts spectral similarity between Q and K using a linear angular kernel. FLattenTransformer (Han et al. (2023)) reduces computation to linear complexity with focused linear attention, effectively maintaining the expressiveness of original attention.

### 2.2 NETWORK PRUNING

Pruning involves identifying and removing redundant components to reduce the network size. Pruning has proven highly effective in original Convolutional Neural Networks (CNNs) (Lin et al. (2020); Hou et al. (2022); Gao et al. (2023); Chen et al. (2023)). Building on the strong performance of CNNs, there has been researches aimed at enhancing the efficiency of Vision Transformers through pruning. One approach, token pruning, involves finding and removing unnecessary tokens, effectively reducing the computational load of the attention mechanism (Bolya et al. (2022); Wei et al. (2023); Tang et al. (2023)). Another approach is to reduce the size of the Transformer model itself. SViTE (Chen et al. (2021b)) determines redundant components via sparse training and prunes accordingly. Unified Visual Transformer Compression (UVC) (Yu et al. (2022b)) reduces network size by combining pruning methods with various compression techniques. Novel ViT (NViT) (Yang et al. (2023)) enables global structural pruning based on a Hessian-based structural pruning criterion. It uses only their heuristic criterion when pruning multi-heads. In contrast, the proposed method takes into account the representation space of each head, minimizing the information loss of the original model's attention mechanism.

To address these issues, there are automatic pruning methods that identify redundant components through training. AutoPrune (Xiao et al. (2019)) determines the necessity of weights based on trainable auxiliary parameters. MetaPruning (Liu et al. (2019)) involves training a meta-network, PruningNet, to find the optimal structure for a given target network. DHP (Li et al. (2020)) applies differentiable hypernetworks to determine the configuration of each channel in the backbone network. Instead of using a hypernetwork, another approach involves a pruning module to each component, learning the importance and removing components (Li et al. (2022); Lee et al. (2024)). VTP (Zhu et al. (2021)) applies automatic pruning to Transformers by using trainable gate variables to identify and remove unnecessary components through learning. Automatic pruning can identify the efficient network structure without heuristics, but directly applying them to Transformers is not trivial. Unlike the traditional CNN structure, an automatic pruning method should take into account the multi-head structure in Transformers without losing expression capacity. In this paper, we propose a multi-head-aware automatic pruning method, which can be applied to both original attention and linear attention mechanisms.

## 3 METHOD

### 3.1 MULTI-HEAD AUTOMATIC PRUNING

In this section, we propose an automatic pruning method for handling multi-head models. It consists of two processes: computing the pruning indicator and adjusting the pruning indicator. Fig. 3(a) illustrates a step of computing pruning indicator, where the importance of each channel is estimated to assign scores. As in previous automatic pruning methods, a pruning indicator consisting of learnable parameters $m \in \mathbb{R}^{C_{out}}$ is employed to determine importance during training. We assign weights to the pruning indicator based on head-wise similarity, considering more informative elements of the representation subspace. First, the projection matrix $P \in \mathbb{R}^{C_{in} \times C_{out}}$ is divided into $P^1, P^2, ..., P^h \in \mathbb{R}^{C_{in} \times C_h}$ based on the number of heads $h$, where $C_{in}, C_{out}$ and $C_h$ are the number of input, output and head channels, respectively. To compute the importance, we first compute

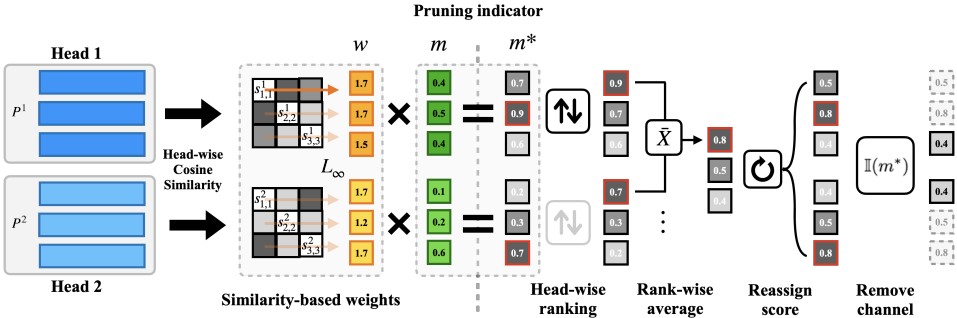

(a) Computing pruning indicator          (b) Pruning indicator adjustment

Figure 3: **Multi-head automatic pruning process.** To consider multi-head, we first perform the (a) Computing pruning indicator process. We integrate similarity-based weights into pruning indicators, allowing them to consider saliency channels in each head. (b) Through Pruning indicator adjustment, we share rank-wise pruning indicators for each head. This ensures equal channel removal across all heads, preventing channel mismatch.

the cosine similarity of the channels for each head as:

$$S^k = \frac{P^k \cdot (P^k)^T}{\|P^k\|\|P^k\|} = \begin{pmatrix} s_{1,1}^k & \cdots & s_{1,C_h}^k \\ \vdots & \ddots & \vdots \\ s_{C_h,1}^k & \cdots & s_{C_h,C_h}^k \end{pmatrix} \tag{1}$$

where $S^k$ is the similarity matrix obtained from the cosine similarity of $P^k$, where element $s_{i,j}^k$ represents the relationship between the $i$-th channel and the $j$-th channel in head $k$. To incorporate this into the pruning indicator $m$, we calculate the similarity-based weights $w \in \mathbb{R}^{C_{out}}$ as follows:

$$w_i^k = 1 + \lim_{p \to \infty} \left( \sum_{n=1,n\neq i}^{C_h} |s_{i,n}^k|^p \right)^{\frac{1}{p}}, \tag{2}$$
$$w = \text{Concat}[w^1, w^2, \cdots, w^h].$$

For each row, we estimate similarity by applying the Chebyshev norm ($L_\infty$) to the remaining elements excluding itself. We apply the Chebyshev norm for two main reasons. First, a well-matched single instance has higher redundancy compared to several less similar ones. Incorporating channel redundancy into the pruning indicator via weighting in Eq. (3) results in Chebyshev norm values less than 1, which ensures more stable training. Through this process, $w_i^k$ represents the similarity between the $i$-th channel and the other channels in head $k$. If $w_i^k$ has a high score for a specific channel, it indicates the presence of other channels with high similarity, implying that it can be replaced by those channels. By combining $w_1^k, w_2^k, \ldots, w_{C_h}^k \in w^k$ using the concatenation operation $\text{Concat}(\cdot)$, we obtain the similarity-based weight $w$. These values are weighted in the pruning indicator as follows:

$$m^* = w \odot m \tag{3}$$

where $\odot$ represents the element-wise product. Fig. 3(b) illustrates the method for adjusting pruning indicators. First, the rank for each head is computed according to $m^*$. The score is calculated by taking the average of $m^*$ with the same rank in each head. This average is then reassigned to the head positions with the same rank, enabling the adjustment of pruning indicators to share the same importance score across all heads. Consequently, redundant channels are masked out by passing through the following indicator function:

$$\mathbb{I}(m^*) = \begin{cases} 1, & \text{if } m^* < \tau \\ 0, & \text{otherwise.} \end{cases} \tag{4}$$

Since the same rank across all heads holds the same pruning indicator, the removal of channels corresponding to each rank is simultaneously determined. This allows training to ensure that pruning occurs at the same proportion across all heads, thereby preventing channel mismatch.

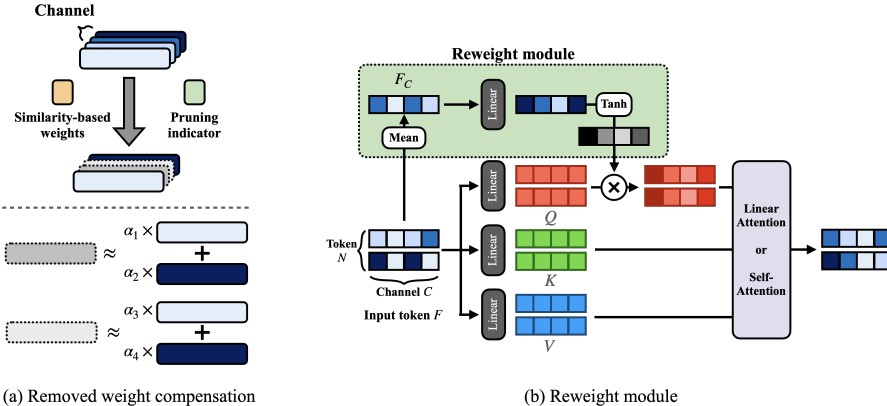

(a) Removed weight compensation      (b) Reweight module

Figure 4: **Reweight module configuration.** (a) The removed channels can be approximated by the weighted sum of the remaining channels. (b) The reweight module consists of a simple channel attention structure, which determines the weights that need to be compensated for each channel.

## 3.2 REWEIGHT MODULE

The removal of channels results in information loss, requiring a method to compensate for this to minimize performance degradation. As in Fig. 4(a), our method learns indicators by considering the similarity of channels for each head, and subsequently removes channels based on these values. The channels being removed can be represented by the weighted sum of the remaining channels, due to their high similarity with other channels.

Fig. 4(b) illustrates the reweight module. The reweighting method employes a simple channel attention mechanism inspired by SENet (Hu et al. (2018)). The input token $F \in \mathbb{R}^{N \times C}$ is compressed into a feature map $F_C \in \mathbb{R}^{1 \times C}$ by taking the mean. The compressed feature map is encoded into weights to compensate for each channel through a linear layer and $Tanh$ activation function. This is multiplied with the query, allowing the passage of information from the removed channels to the remaining channels. By applying this module, compensation can be provided for the removed channels, minimizing performance degradation.

## 3.3 PRUNING INDICATOR INITIALIZATION FOR LINEAR ATTENTION

Unlike original attention in Transformer, linear attention with a computational complexity of $\mathcal{O}(NC^2)$, significantly improves in efficiency as channels are removed. Since the channels of linear attention are likely to be severely removed during the pruning process, it is non-trivial to properly determine an initial constant value to prevent it. Therefore, a sophisticated initialization of the pruning indicator is required. We introduce a data-driven method to solve the problem. When the embedding vector is received as input, it is projected into query $Q \in \mathbb{R}^{N \times C}$, key $K \in \mathbb{R}^{N \times C}$, and value $V \in \mathbb{R}^{N \times C}$. The relationship between each token $QK^T$ can be expressed as a linear combination of each channel as shown below.

$$QK^T V = (Q_1 K_1^T + Q_2 K_2^T + \cdots + Q_C K_C^T)V. \tag{5}$$

The importance of each channel $Q_i K_i^T V$ can be determined by calculating the matrix distance from $QK^T V$ in Eq. (5). By projecting $Q_i K_i^T$ onto $V$, we compare matrices in $\mathbb{R}^{N \times C}$ instead of $\mathbb{R}^{N \times N}$, significantly reducing computational complexity when $N \gg C$. We use the difference in singular values to measure the matrix distance. Let $\Sigma^j$ and $\Sigma_i^j$ be the singular values of the $QK^T V$ and the $Q_i K_i^T V$ in $j$-th image of training database, respectively. The importance score of channel i is given by:

$$T_i^j = |\Sigma^j - \Sigma_i^j|. \tag{6}$$

The pruning indicator is initialized by accumulating the distance differences for each image and normalizing them between 0 and 1. Using this initialization, we can retain a sufficient number of channels, ensuring the attention mechanism operates effectively.

### 3.4 TRAINING AUTOMATIC PRUNING METHOD

In the $l$-th layer of the Transformer, the pruning module $m^{*l}$ is applied to the original model's weight $o^l \in \mathbb{R}^{C_{in} \times C_{out}}$ as follows:

$$\tilde{o}_i^l = (\mathbb{I}(m^{*l}_i) \odot o_i^l). \tag{7}$$

where $o_i^l$ and $\tilde{o}_i^l$ denote the weights of the $i$-th channel in the original and pruned layers, respectively. $\mathbb{I}(\cdot)$ is the indicator function in Eq. (4). Since $\mathbb{I}(\cdot)$ is a non-differentiable binary operation, the straight-through estimator (STE) is applied in back-propagation, allowing the pruning indicator to be learned by directly passing the gradient from $\mathbb{I}(m_i^l)$ to $m^{*l}_i$.

The loss for the automatic pruning method is as follows:

$$\mathcal{L} = \mathcal{L}_{CE}(f(x), y) + \mathcal{L}_{FLOPs}(M_{prune}, M_{target}) \tag{8}$$

where $L_{CE}$ is the cross-entropy loss, comparing the model output $f(x)$ for input $x$ with the true label $y$. FLOPs aware loss $\mathcal{L}_{FLOPs}$ is the Euclidean distance between the current pruned model's FLOPs, $M_{prune}$ and the target FLOPs, $M_{target}$. To compute the current FLOPs, the following formula is used:

$$M_{prune} = \sum_{l \in P} N \times \tilde{C}_{in} \times \tilde{C}_{out} + \sum_{l \in LA} (N \times \tilde{C}_q \times \tilde{C}_k + N \times \tilde{C}_k \times \tilde{C}_v) + \sum_{l \in OA} (N^2 \times \tilde{C}_q + N^2 \times \tilde{C}_v) \tag{9}$$

where $P$ denotes the projection matrix layer, and $\tilde{C}_{in}, \tilde{C}_{out}$, and $N$ represent the numbers of remaining input and output channels and tokens, respectively. Unlike traditional CNNs, Transformers have additional computational costs due to the attention mechanism. $LA$ and $OA$ represent linear attention and original attention, respectively, with $\tilde{C}_q, \tilde{C}_k$, and $\tilde{C}_v$ indicating the remaining numbers of query, key, and value channels. Considering the reconfiguration process, the pruning indicator $m^*$ for query and key projection is shared to ensure they are pruned at the same proportion.

## 4 EXPERIMENTS

To verify the proposed method, we apply our Automatic Multi-head Attention Pruning (AMAP) to the FLattenTransformer (Han et al. (2023)). This network is composed of original attention blocks from the SwinTransformer (Liu et al. (2021)) and linear attention blocks. Unlike previous pruning methods, the proposed approach can be applied to both types of attentions without specific heuristics. The largest model, AMAP-L, is a compressed model of FLatten-Swin-B, with a size of 11.1G. AMAP-B compresses the FLatten-Swin-S model to 6.2G, while AMAP-S and AMAP-T compress the FLatten-Swin-T model to 4.2G and 1.3G, respectively. While other models follow the hybrid structure of FLattenTransformer, the smallest model, AMAP-T, consists solely of linear attention to achieve a higher compression ratio. We evaluate the performance of the compressed models on the classification task using the ImageNet-1K dataset (Deng et al. (2009)). To compress the network, we employ a search and refine process. For more detailed information on the training, please refer to Appendix A.

### 4.1 COMPARISON WITH PREVIOUS METHODS

In Table 1, we compare our AMAP models with previous efficient models. Compared to vision transformer variants, our method demonstrates superior performance with lower computational cost. For example, AMAP-L outperforms Swin-B and AS-ViT-L (Chen et al. (2022)) with 30% and 50% fewer FLOPs, respectively. Notably, compared to DeiT-B (Touvron et al. (2021a)), it achieves a 1.96% performance gain with 37% fewer FLOPs. AMAP-S achieves 1.05% better performance than Swin-T (Liu et al. (2021)) while requiring approximately 10% fewer computations. The most efficient model, AMAP-T, surpasses DeiT-T by 4.33% and outperforms GLiT-T (Chen et al. (2021a)) with similar computational resources.

We also compare ours with efficient networks using pruning method. Our proposed method surpasses NViT (Yang et al. (2023)), a human knowledge-based pruning approach, by up to 0.32% at the same FLOPs. AMAP-L outperforms EviT (Liang et al. (2022)) by 1.66% with fewer FLOPs and achieves up to 0.24% improvement over the state-of-the-art DIMAP (He & Zhou (2024)) with approximately 20% fewer FLOPs. Furthermore, APMA-B surpasses OFB (Ye et al. (2024)) by 1.49%

| | Venue | Acc. (%) ↑ | FLOPs (G) ↓ |
|---|---|---|---|
| DeiT-T | ICML 2021 | 72.2 | 1.3 |
| DeiT-S | Touvron et al. (2021a) | 79.8 | 4.6 |
| DeiT-B | | 81.8 | 17.5 |
| T2T-ViT-t-14 | ICCV 2021 | 81.7 | 6.1 |
| T2T-ViT-t-19 | Yuan et al. (2021) | 82.4 | 9.8 |
| T2T-ViT-t-24 | | 82.6 | 15 |
| CaiT-XXS-24 | ICCV 2021 | 77.6 | 2.5 |
| CaiT-XS-24 | Touvron et al. (2021b) | 81.8 | 5.4 |
| CaiT-S-24 | | 82.7 | 9.4 |
| CvT-13 | ICCV 2021 | 81.6 | 4.5 |
| CvT-21 | Wu et al. (2021) | 82.5 | 7.1 |
| GLiT-T | ICCV 2021 | 76.3 | 1.4 |
| GLiT-S | Chen et al. (2021a) | 80.5 | 4.4 |
| GLiT-B | | 82.3 | 17 |
| Swin-T | ICCV 2021 | 81.3 | 4.5 |
| Swin-S | Liu et al. (2021) | 83 | 8.7 |
| Swin-B | | 83.5 | 15.4 |
| AS-ViT-S | ICLR 2022 | 81.2 | 5.3 |
| AS-ViT-B | Chen et al. (2022) | 82.5 | 8.9 |
| AS-ViT-L | | 83.5 | 22.6 |
| STViT-Swin-T | CVPR 2023 | 81.0 | 3.1 |
| STViT-Swin-S | Chang et al. (2023) | 82.8 | 5.9 |
| STViT-Swin-B | | 83.2 | 10.48 |
| Flatten-Swin-T | ICCV 2023 | 82.1 | 4.5 |
| Flatten-Swin-S | Han et al. (2023) | 83.5 | 8.7 |
| Flatten-Swin-B | | 83.8 | 15.4 |

(a) Vision transformer variants

| | Venue | Acc. (%) ↑ | FLOPs (G) ↓ |
|---|---|---|---|
| EViT-DeiT-S | ICLR 2022 | 81.3 | 3 |
| EViT-DeiT-B | Liang et al. (2022) | 82.1 | 11.6 |
| UVC-DeiT-S | ICLR 2022 | 78.82 | 2.3 |
| UVC-DeiT-B | Yu et al. (2022b) | 80.57 | 8 |
| WDPruning | AAAI 2022 | 81.8 | 6.3 |
| WDPruning | Yu et al. (2022a) | 82.2 | 6.8 |
| WDPruning | | 82.41 | 7.6 |
| NViT-T | CVPR 2023 | 76.21 | 1.3 |
| NViT-S | Yang et al. (2023) | 82.19 | 4.2 |
| NViT-B | | 82.95 | 6.2 |
| X-Pruner | CVPR 2023 | 80.7 | 3.2 |
| X-Pruner | Yu & Xiang (2023) | 82 | 6 |
| SPViT | TPAMI 2024 | 82.4 | 8.4 |
| SPViT-Swin-S | He et al. (2024) | 82.4 | 6.1 |
| OFB-Swin-T | CVPR 2024 | 79.9 | 2.6 |
| OFB-DeiT-B | Ye et al. (2024) | 80.3 | 3.6 |
| OFB-DeiT-B | | 81.7 | 8.7 |
| Swin-T-DIMAP1 | ICLR 2024 | 81.71 | 3.8 |
| Swin-S-DIMAP1 | He & Zhou (2024) | 83.08 | 7.5 |
| Swin-B-DIMAP1 | | 83.52 | 13.2 |
| **AMAP-Swin-T** | | **81.96** | **4.2** |
| **AMAP-Swin-S** | | **82.72** | **6.2** |
| **AMAP-T** | | **76.53** | **1.3** |
| **AMAP-S** | | **82.35** | **4.2** |
| **AMAP-B** | | **83.19** | **6.2** |
| **AMAP-L** | | **83.76** | **11.1** |

(b) Efficient networks using pruning methods

Table 1: **ImageNet-1K results** of various efficient models and our method, AMAP. Our approach demonstrates higher performance at lower FLOPs than both vision transformer variants and efficient networks using pruning methods.

| | Acc. (%) ↑ | FLOPs (G) ↓ | Throughput ↑ | Speed up |
|---|---|---|---|---|
| FLatten-Swin-B Han et al. (2023) | 83.8 | 15.4 | 340 | x1 |
| AMAP-L | 83.76 (-0.04) | 11.1 (**30%** ↓) | 463 | x 1.4 |
| FLatten-Swin-S | 83.5 | 8.7 | 492 | x1 |
| AMAP-B | 83.19 (-0.31) | 6.2 (**30%** ↓) | 663 | x 1.3 |
| FLatten-Swin-T | 82.1 | 4.5 | 805 | x 1 |
| AMAP-S | 82.35 (+0.25) | 4.2 (10% ↓) | 921 | x 1.1 |
| AMAP-T | 76.53 (-5.57) | 1.3 (70% ↓) | 1221 | x 1.5 |

Table 2: **Comparison between FlattenTransformer and AMAP.** All measurements are conducted under the same computational environment on RTX A6000. Note that AMAP-S achieves better throughput compared to Flatten-Swin-T, while showing better accuracy.

with 30% fewer FLOPs and obtains performance gains of 1.39% and 1.19% over WDPruning (Yu et al. (2022a)) and X-Prune (Yu & Xiang (2023)), respectively, with similar computational cost. While other networks rely on original attention mechanisms with a computational complexity of $\mathcal{O}(N^2C)$, our method also compresses linear attention-based networks, which have a complexity of $\mathcal{O}(NC^2)$. As a result, even with less channel pruning, it is possible to achieve the same reduction in computational cost, thereby maintaining a greater expressive capacity.

To demonstrate the general applicability of our method, we apply it to SwinTransformer and compare it with other pruning techniques. AMAP-Swin-T and AMAP-Swin-S are compressed versions of Swin-T and Swin-S, respectively. With nearly the same FLOPs, AMAP-Swin-S, which compresses the original model by 30%, surpasses the previous state-of-the-art X-Pruner (Yu & Xiang (2023)) by 0.72% and SPViT (He et al. (2024)) by 0.32%. AMAP-Swin-T is more efficient and achieves around 0.66% better performance compared to Swin-T. These results show that the proposed method can be effectively applied to multi-head attention networks.

We also compare the performance of the state-of-the-art FLattenTransformer (Han et al. (2023)) backbone with our proposed method applied. AMAP-L and AMAP-B enable near-lossless compression of the FLatten-Swin-B and FLatten-Swin-S models, respectively, reducing computational cost about 30% with only a 0.04% and 0.31% performance drop. As shown in Table 2, this leads to a throughput improvement of around 1.3x to 1.4x. AMAP-S achieves a performance gain of 0.25% over the FLatten-Swin-T model while also reducing computational cost and increasing computation speed, demonstrating the effectiveness of our method in optimizing model performance.

## 4.2 ABLATION STUDY

In this section, we show the effectiveness of each module and method by individually removing them and comparing the performance. All experiments, except the one in Table 3(b), are performed using

|  | Acc. (%) ↑ | FLOPs (G) ↓ |
|---|---|---|
| Original | 76.53 | 1.3 |
| (-) Reweight module | 76.36 (-0.17) | 1.3 |
| (-) Similarity-based weight | 75.90 (-0.63) | 1.3 |
| (-) Multi-head automatic pruning | 72.78 (-3.75) | 1.3 |

(a) Impact of each module

|  | Acc. (%) ↑ | FLOPs (G) ↓ |
|---|---|---|
| AMAP-B | 83.19 | 6.2 |
| AMAP-B † | 82.72 (**-0.47**) | 6.2 |
| AMAP-S | 82.35 | 4.5 |
| AMAP-S † | 82.17 (**-0.18**) | 4.5 |
| AMAP-T | 76.53 | 1.3 |
| AMAP-T † | 71.04 (**-5.49**) | 1.3 |

(b) Impact of pruning indicator initialization

Table 3: **Evaluation of the effectiveness of proposed methods.** † denotes no initialization. Significant performance degradation is observed when each module and method is removed.

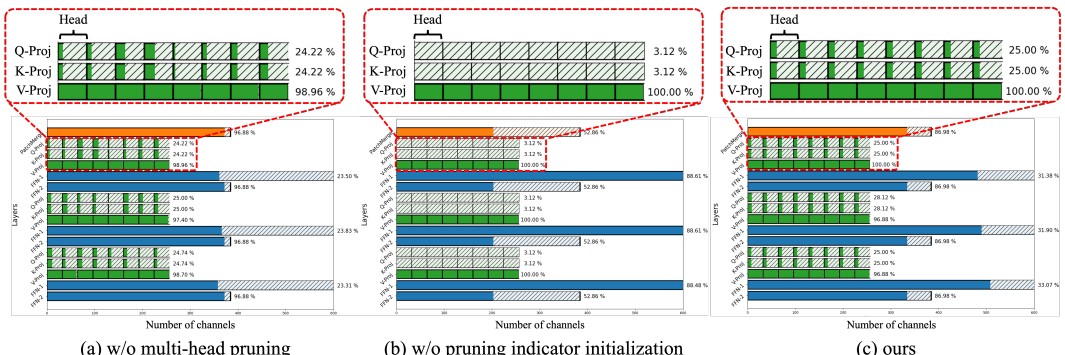

(a) w/o multi-head pruning  (b) w/o pruning indicator initialization  (c) ours

Figure 5: **Pruned model structure after reconfiguration.** (a) Without applying multi-head pruning, the pruning ratio for each head is inconsistent, leading to a channel mismatch problem. (b) Without indicator initialization for linear attention, the attention mechanism does not function properly. (c) Our proposed method ensures consistent pruning ratios across heads to resolve the channel mismatch problem and demonstrates effective pruning of each module in appropriate proportions.

AMAP-T, and the experimental setup is consistently configured. The similarity-based weight and multi-head automatic pruning are proposed in Sec. 3.1, reweight module in Sec. 3.2, and pruning indicator initialization in Sec. 3.3.

**Multi-head Automatic Pruning** In contrast to conventional automatic pruning methods, our proposed approach can remove channels considering the multi-head structure. Without considering multi-heads, the accuracy drops by **3.75%**, indicating a significant performance drop, as shown in Table 3 (a). In Fig. 5(a), the channels for query, key, and value vary across each head. During the reconstruction process in the network, where redundant channels are actually removed for real acceleration, channel mismatch occurs across heads. The almost complete removal of channels in specific heads leads to restricted representation space and severe performance degradation. On the other hand, applying our proposed method, as shown in Fig. 5(c), ensures that channels are removed at the same proportion across heads, thereby preserving the network's expression capacity.

**Similarity-based Weight** We analyze the singular value norms of each attention layer in the model right after the search process. Given the varying number of channels in the model with and without the similarity-based weight, we calculate the norm of the top three singular values for each head. Fig. 6 shows that the singular value norms are higher across all layers when the similarity-based weight is applied. This indicates that the model has eigenvectors of greater significance, enabling it to inherently extract more distinct features. Consequently, as shown in Table 3(a), the performance gap between the model using the similarity-based weight and the model without it is a significant 0.63%.

**Reweight Module** We apply a reweight module to compensate for the information loss that occurs when channels are removed. As shown in Table 3 (a), when the module is removed in the proposed automatic pruning method, there is a decrease in accuracy of 0.17% without any change in FLOPs. Fig. 7 illustrates the attention for tokens with and without applying the module. When applying the reweight module, the relationship with relevant object tokens strengthens, similar to the original model, while the relationship with background tokens weakens. The result demonstrates that the reweight module allows relevant tokens to receive better attention.

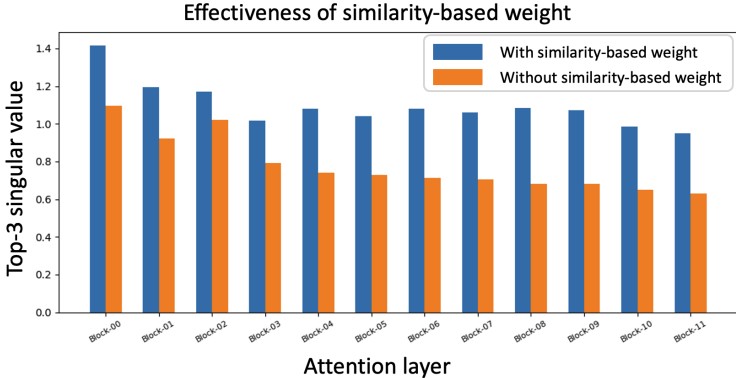

Figure 6: **Top-3 singular value norms** for each attention layer of the pruned model. When using similarity-based weights, it demonstrates larger singular value norms. It indicates that when employing the proposed method, salient channels can be effectively retained.

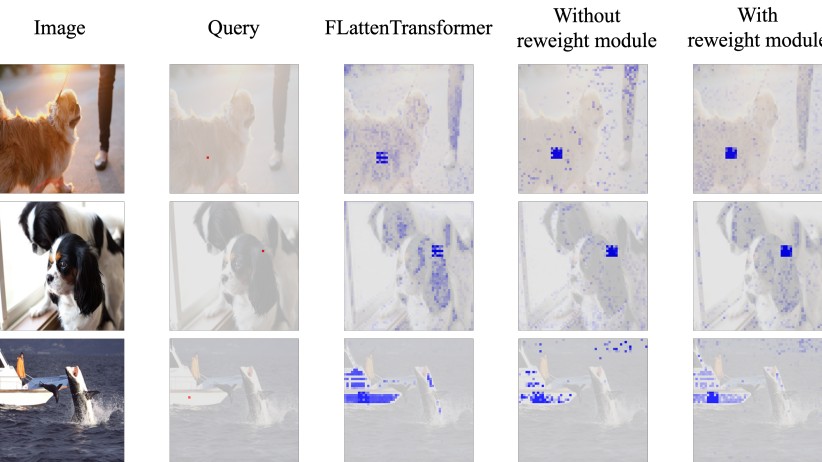

Figure 7: **The effect of the reweight module.** The red block represents the query token, while the blue blocks depict the relationship between this query token and other tokens. It demonstrates that the reweight module can compensate for information loss caused by pruning.

**Pruning Indicator Initialization for Linear Attention**   Fig. 5(b) shows the model structure obtained from the search process when the pruning indicator for linear attention is not initialized. It illustrates that channels of the projection matrix for the query and key are severely pruned, whereas the feed-forward network is not pruned. This imbalance leads to improper operation of the attention mechanism. In contrast, Fig. 5(c) demonstrates how our method alleviates such issues. Table 3 (b) presents the impact of pruning indicator initialization across various sizes of our models. In the highly compressed AMAP-T model, without pruning indicator initialization, there is a significant performance degradation of 5.49%. For AMAP-B and AMAP-S, the performance drops by 0.47% and 0.18%, respectively, showing less impact compared to the model with higher compression. However, there is still observable performance improvement, demonstrating the effectiveness of our proposed initialization method.

## 5   CONCLUSION

In this paper, we introduce an AMAP (*Automatic* Multi-head Attention Pruning) method. Integrating similarity weights into the trainable scheme allows us to progressively achieve a more optimal structure compared to other pruning methods that rely on deterministic metrics. Through an ablation study, we validate the impact of the proposed similarity-based pruning indicator, reweight module, and initialization method. Comparative analysis against vision transformer variants and previous pruning methods demonstrates the superior efficiency and performance trade-off of AMAP.

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

## A   TRAINING DETAILS

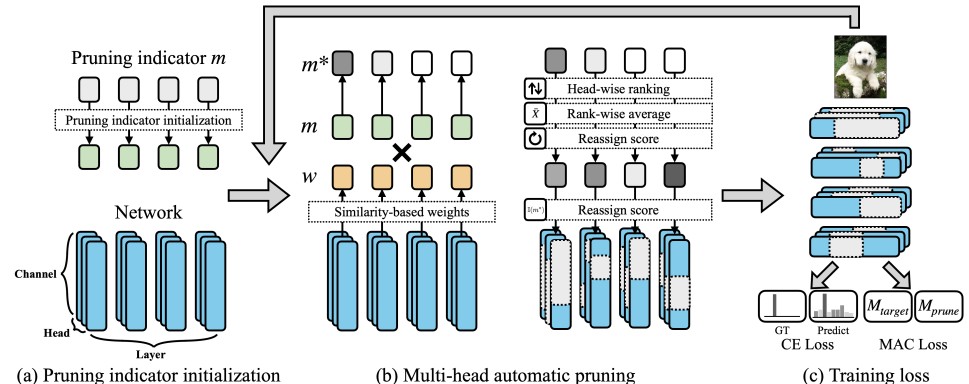

(a) Pruning indicator initialization     (b) Multi-head automatic pruning     (c) Training loss

Figure 8: Search pipeline. To perform network search, (a) we initialize the pruning indicator, (b) apply multi-head automatic pruning to mask unnecessary channels, and (c) train the network using cross-entropy loss and FLOPs loss.

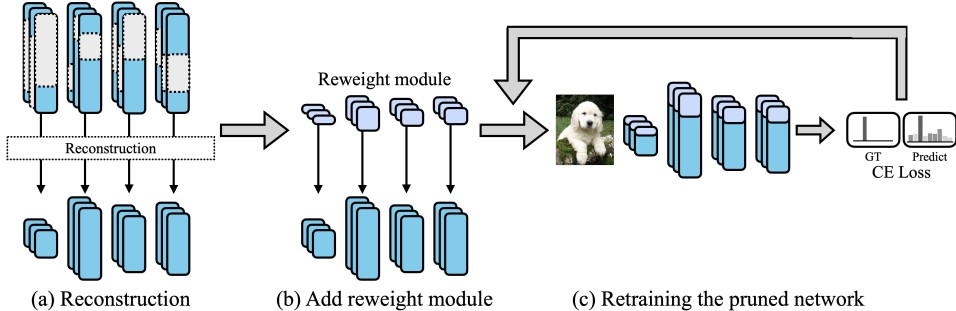

(a) Reconstruction     (b) Add reweight module     (c) Retraining the pruned network

Figure 9: Refine pipeline. To accelerate the retraining process, (a) we reconstruct the network and (b) add a reweight module to compensate for information loss. Then, (c) we retrain the pruned network.

To compress the network, we perform search and refine process. The search process involves determining the efficient network through an automatic pruning module over 30 epochs. We use the AdamW optimizer, starting with a learning rate of 5e-4 and decaying to 5e-6, with a weight decay set to 1e-6. The batch size is 1024, and the training is conducted on 8 RTX A6000 GPUs. The refine process aims to recover any information loss during the search phase. Except for setting the weight decay, the experimental setup remains the same as in the search process. AMAP-L, AMAP-B and AMAP-S apply a weight decay of 0.05, while AMAP-T uses a weight decay of 1e-6 to minimize additional sparsity. In all experiments, the threshold $\tau$ for the indicator function is set to 0.5.

The proposed method for compressing the network involves a two-step process: a search to find the efficient structure and a refinement to restore the loss information. Fig. 8 illustrates the search pipeline. First, the pruning indicator is initialized (Sec. 3.3). Then, the proposed similarity-based weights are computed, and an automatic pruning method is applied to reassign the scores for all heads using the pruning indicator, masking unnecessary channels (Sec. 3.1). During the search process, cross-entropy loss for classification and FLOPs loss to achieve the target compression rate are used to train the network (Sec. 3.4).

After the search process, the refinement procedure to restore network performance is shown in Fig.9. To accelerate retraining, the network is reconstructed based on the structure obtained during the search process. Reweight modules are added to compensate for the information loss caused by pruning (Sec. 3.2). Subsequently, the network is trained using cross-entropy loss.

Table A shows the time required for the Search and Refine processes. The experiments are conducted using the Intel(R) Xeon(R) Gold 6226R CPU @ 2.90GHz for the CPU and the NVIDIA RTX A6000

| | Search time (hours) | Refine time (hours) |
|---|---|---|
| AMAP-L | 17 | 110 |
| AMAP-B | 10 | 69 |
| AMAP-S | 7 | 48 |
| AMAP-T | 7 | 42 |

Table 4: **The computing hours** for the search and refine processes.

49GB for the GPU. The largest model, AMAP-L, requires 17 hours for the search process and 110 hours for the refine process. AMAP-B requires 12 hours for model search, and an additional 72 hours for the refine process to recover information loss. Since AMAP-S and AMAP-T prune FLatten-Swin-T, their search time is the same at 7 hours. After pruning, AMAP-S and AMAP-T require 48 hours and 42 hours, respectively, in the refine process.

