# OpenReview forum: "AMAP: Automatic Multi-head Attention Pruning by similarity-based pruning indicator"
_ICLR.cc/2025/Conference — ICLR 2025 Conference Withdrawn Submission_

### Official Review · Reviewer_k7c9 · 2024-10-27

**Soundness:** 3
**Presentation:** 3
**Contribution:** 3
**Rating:** 5
**Confidence:** 3

**Summary:**

This paper develops an Automatic Multi-head Attention Pruning (AMAP) method for reducing the computational complexity of transformers. The key idea is to integrate channel similarity-based weights into the pruning indicator to address a "channel mismatch problem" (as defined by the authors). The proposed idea is sensible and supported by experimental results. Around 1-2% performance improvement over Swin Transformer has been achieved with the reduction of FLOPs (30-37%).

**Strengths:**

+This paper is well written and easy to follow. The motivation is the "channel mismatch problem" as described in the introduction. A multi-head pruning process is introduced in Fig. 3 to prevent the problem.
+ The reported experimental results are comprehensive and convincing. Table 1 compares the proposed AMAP with current SOTA (including CVPR'2024 and ICLR'2024). The results show AMAP can achieve higher accuracy with a smaller amount of FLOPs.

**Weaknesses:**

-Originality: I am having a hard time appreciating the novelty of the proposed approach. In the main body (Sec. 3), Sec. 1 and 2 seem standard procedure. The main contributions, if I am correct, lie in Sec. 3.3 and 3.4. In Sec. 3.3, "We introduce a data-driven method to solve the problem" - without a single reference, I am wondering how is the proposed pruning indicator related to the existing literature. The idea presented in Eqs. (5) and (6) seem straightforward. In Sec. 3.4, I am not sure if the straight-through estimator (STE) is the authors' new contribution or cited from the literature.
-Clarity: I am a bit confused by the use of "automatic pruning" in the title, especially when I see the loss function in Eq. (8). In my biased opinion, a method can not be named "automatic" if the objective function involves the performance metric itself (i.e., FLOPs). I have seen many other pruning techniques without involving FLOP-aware loss during training (but only during testing). I could be wrong - but will Eq. (9) cause some kind of chicken-and-egg problem? Is it true that the lower the better (for FLOPs)?
-Significance: After reviewing many pruning-related paper, I have found this paper has made incremental contribution to the field. "Similarity-based pruning indicator" is a sensible idea but I am not convinced about the significance along this line of research. For example, how about the generalization property of AMAP? Does it work on different datasets and architectures?

**Questions:**

1) Token pruning only represents one possible attack to the complexity-performance tradeoff in ViT. What about other strategies such as token fusion? For example, Multi-criteria Token Fusion (MCTF) was proposed in CVPR'2024 (https://github.com/mlvlab/MCTF). I am wondering if the authors have considered the possibility of combining pruning with fusion in their design?
2) Have you studied the limitation of cosine similarity in Eq. (1)? What about other alternatives such as Pearson correlation or Euclidean distance?
3) What studies have you done to understand the generalization of similarity-based pruning indicators? Will the defined "Channel mismatch problem" become more pronounced for multimodal setting such as vision-language model (VLM)? I think the significance of this work can benefit a lot if it can be generalized to VLM.

---

### Official Review · Reviewer_znfx · 2024-10-31

**Soundness:** 2
**Presentation:** 2
**Contribution:** 3
**Rating:** 5
**Confidence:** 4

**Summary:**

In this paper, the authors propose an automatic pruning method to address the channel mismatch issue. They introduce channel similarity-based weights into the pruning indicator to preserve information and mitigate channel mismatch. Multiple experiments indicate that the proposed method can effectively reduces FLOPs with minimal accuracy drop.

**Strengths:**

1. The experiment parts is well written. The experimental results, particularly on ImageNet-1K, demonstrate the method’s efficacy across multiple model configurations.

2. The method tackles the channel mismatch problem inherent in multi-head pruning, which is well illustrated with diagrams. This multi-head awareness in pruning is particularly relevant for improving model efficiency in real-world applications where memory and computation are limited.

3. The paper is well structured and easy to understand.

**Weaknesses:**

1. In Figure 2, the paper demonstrates that the proposed AMAP eliminate redundant channels within each head, highlighting the benefits  over previous methods which may remove unbalanced channels from each head. However, the authors do not elaborate on why this choice specifically benefits multi-head architectures or how it may differ from other pruning methods in terms of learned visual representations. Make these factors clear may makes the method more convincing.

2. The primary contribution of this paper is its efficiency improvements. However, while detailed theoretical complexity metrics (e.g., FLOPs) are provided, on-device speed metrics are limited, appearing only in Table 2. Given that the benchmark implementation exists, could the authors clarify why on-device speed measurements were not included in the remaining ablation studies and comparison experiments? Adding these metrics would strengthen the practical relevance of the reported findings.

3. This paper reports only on-GPU speed (e.g., Throughputs in Table 2). To better substantiate the efficiency contribution, please consider adding speed comparisons on CPU and mobile devices using off-the-shelf inference frameworks, such as TNN. These additional metrics would provide a more comprehensive view of the model's efficiency across various deployment scenarios (like many existing works [1,2]).

4. The comparison methods in Table 1 are not particularly strong in terms of efficiency. To better support the claimed contribution, please consider including some state-of-the-art efficiency-focused methods, such as FastViT [2], SwiftFormer [3], and MobileOne [4]. Additionally, implementing on-device speed comparisons (e.g., GPU or CPU latency/throughputs) would provide a clearer and more robust demonstration of the efficiency advantages of the proposed method.

5.  The paper lacks an in-depth discussion on the impact of different pruning ratios, raising questions about the boundaries of these ratios and their effects on model performance. A more detailed analysis of varying pruning ratios would clarify the method’s sensitivity to different pruning intensities, helping to establish optimal settings and offering insights into the trade-offs between computational efficiency and model accuracy.

6. To enhance the persuasiveness of the experimental section, it would be beneficial to include intuitive performance comparisons between channel pruning and token pruning methods. This addition would provide clearer insights into the relative strengths and limitations of each approach, reinforcing the practical advantages of the proposed method.

7. Since the authors remove redundant channels in each head, I wonder how the number of heads affects the accuracy of the proposed method. The information capacity may change with different channel numbers, and understanding this relationship could provide deeper insights.

In summary, this paper is well-motivated and presents a sufficient level of novelty. However, the insufficient experimental and explanatory contents weaken the overall contribution. I would be pleased to re-evaluate the paper once the necessary experiments and improvements in explanation have been addressed.

[1] Chen J, Kao S, He H, et al. Run, don't walk: chasing higher FLOPS for faster neural networks[C]//Proceedings of the IEEE/CVF conference on computer vision and pattern recognition. 2023: 12021-12031.

[2] Vasu P K A, Gabriel J, Zhu J, et al. FastViT: A fast hybrid vision transformer using structural reparameterization[C]//Proceedings of the IEEE/CVF International Conference on Computer Vision. 2023: 5785-5795.

[3] Shaker A, Maaz M, Rasheed H, et al. Swiftformer: Efficient additive attention for transformer-based real-time mobile vision applications[C]//Proceedings of the IEEE/CVF International Conference on Computer Vision. 2023: 17425-17436.

[4] Vasu P K A, Gabriel J, Zhu J, et al. Mobileone: An improved one millisecond mobile backbone[C]//Proceedings of the IEEE/CVF conference on computer vision and pattern recognition. 2023: 7907-7917.

**Questions:**

Please deal with the major problems above. There are some minor issues above.

1. In Line 130, the "achieveing" should be "achieving."

2. It would be better to present some limitations on the CONCLUSION section.

---

### Official Review · Reviewer_6knJ · 2024-11-03

**Soundness:** 3
**Presentation:** 3
**Contribution:** 2
**Rating:** 5
**Confidence:** 3

**Summary:**

This paper introduces an AMAP (Automatic Multi-head Attention Pruning) method. Integrating similarity weights into the trainable scheme allows us to progressively achieve a more optimal structure compared to other pruning methods that rely on deterministic metrics. The work presents a range of experiments that sufficiently support its claims. It is very interesting for readers.

Overall, it is a good read. The manuscript might get better if a few suggestions (given below) are incorporated.

**Strengths:**

1. The writing is easy to read and clearly explains everything in the paper.
2. The experimental result is good compared to the previous works. Empirically, the method seems to offer strong accuracy, compared to existing methods with similar architectures.

**Weaknesses:**

1. The related work is comprehensive. However, the authors only highlight the salient features of the previous works that they apply in their network. The manuscript can benefit from discussing shortcomings of the existing methods as research gaps in the section "Related Work".
2. The expression of the Eq.(6) is ambiguous， especially the expression of Σ. Readers are hard to understand.
3. In Eq.(8), please write M_target.
4. Author should write the pruning process in details in Section 3.3.

**Questions:**

1. The related work is comprehensive. However, the authors only highlight the salient features of the previous works that they apply in their network. The manuscript can benefit from discussing shortcomings of the existing methods as research gaps in the section "Related Work".
2. The expression of the Eq.(6) is ambiguous， especially the expression of Σ. Readers are hard to understand.
3. In Eq.(8), please write M_target.
4. Author should write the pruning process in details in Section 3.3.

---

### Official Review · Reviewer_VHas · 2024-11-04

**Soundness:** 2
**Presentation:** 3
**Contribution:** 3
**Rating:** 5
**Confidence:** 4

**Summary:**

This article mainly discusses how to prune transformers to reduce the amount of model calculation while maintaining recognition accuracy. This direction is an important research direction and has good application prospects. The method in this article is mainly optimized for Multi-head situations, and AMAP (Automatic Multi-head Attention Pruing) is proposed.  For the implementation part, The compression is based on FLatternTransformer (SwinTransformer and linear attetion blocks). Finally, experiments were conducted on ImageNet-1K, and the proposed method achieved a certain improvement over previous methods.

**Strengths:**

1. The research direction covered in the article is a good topic, and is of great reseach value.
2.The overall organization and writing of the entire article are well done, especially in Figure 2. It effectively illustrates the basic idea of the proposed method, as well as the differences from existing works and the theoretical improvements.
3.The approach has some level of innovation, but it may not be particularly outstanding. The overall framework, which involves threshold truncation and fine-tuning, bears similarities to early CNN pruning techniques. But anyway, it can be considered a notable breakthrough if this is the first application of this technique to transformers.
4.In addition to analyzing the theoretical GFLOPS, it is also important to test the acceleration ratio on specific hardware. Evaluating the effectiveness of the method through actual runtime testing is crucial.

**Weaknesses:**

1. The experiments were conducted only on one type of transformer structure, and there is a lack of evaluation on the other transformer structures mentioned in the related work.Additionally, the experiments were only conducted on vision task (ImageNet-1K dataset). However, as transformers are known to be more suitable for language-related tasks, the effectiveness of AMAP has not been demonstrated in the field of natural language processing where transformers are widely used. In other words, it is more suitable for a CV conference instead of a machine learning conference according to current contents.
2.The improvement achieved by the proposed method is relatively limited compared to the method presented in CVPR2023 NVit. In cases where FLOPs are the same, at 1.3G, 4.2G, and 6.2G, the average accuracy improvement is around 0.3%.
3.About the presentation of the results comparison. Using a line graph to visualize the results, particularly for Table 1, which involves multiple methods and model sizes, would provide a clearer and more intuitive comparison. Additionally, including an upper bound reference line would further enhance the effectiveness of the comparison.
4.Some minor issues. Many of the numbers with smaller fonts in Figure 5 are difficult to discern in the printed version. In Figure 3, the white text on a gray background is also hard to distinguish in the print version.

**Questions:**

it would be helpful to double-check the accuracy of the calculated value of m* in Figure 3, as it does not seem to correspond correctly to the description in Equation (3).

---

### Note · Authors · 2024-11-20

**Comment:**

We appreciate the reviewers' comprehensive and detailed comments. We will try to improve our research further.

**Withdrawal Confirmation:**

I have read and agree with the venue's withdrawal policy on behalf of myself and my co-authors.